# Comparison of Outcomes of Silicone Tube Intubation with or without Dacryoendoscopy for the Treatment of Congenital Nasolacrimal Duct Obstruction

**DOI:** 10.3390/jcm12237370

**Published:** 2023-11-28

**Authors:** Doah Kim, Helen Lew

**Affiliations:** Department of Ophthalmology, CHA Bundang Medical Center, CHA University, Seongnam 13496, Republic of Korea; a216015@chamc.co.kr

**Keywords:** congenital nasolacrimal duct obstruction, dacryoendoscpy, silicone tube intubation

## Abstract

In this retrospective study, we compared and analyzed two groups of patients who underwent silicone tube intubation (STI) to treat congenital nasolacrimal duct obstruction (CNDO). We employed dacryoendoscopy to visualize the lacrimal pathways of one group. In total, 85 eyes of 69 patients were included (52 of 41 patients in the non-dacryoendoscopy and 33 eyes of 28 patients in the dacryoendoscopy group). Clinical characteristics, dacryoendoscopic findings, and surgical outcomes were evaluated. The overall STI success rate was 91.8%, and the success rate was significantly higher in the dacryoendoscopy versus non-dacryoendoscopy group (97.0% and 88.5%, respectively). For patients < 36 months of age, the success rate was 100% (23 eyes). All patients with Hasner valve membranous obstructions were younger than 36 months and had structural obstructions of the lacrimal drainage system (LDS) (*p* = 0.04). However, in patients lacking Hasner valve obstructions, LDS secretory (50.0%) and structural (50%) obstructions occurred at similar rates, which did not vary by age. Dacryoendoscopy-assisted STI enhanced the therapeutic efficacy of CNDO and identified diverse CNDO etiologies beyond Hasner valve obstructions. These findings emphasize the potential advantages of dacryoendoscopy in surgical treatment for CNDO patients.

## 1. Introduction

Congenital nasolacrimal duct obstruction (CNDO) is the most common cause of childhood epiphora [1] and is typically caused by obstruction of the Hasner valve at the end of the nasolacrimal duct [2,3]. The spontaneous CNDO resolution rate in the first year of life is in the range of 62.8–95.0% [4,5,6,7,8]. CNDO treatment is stepwise in nature [9]. In infants younger than 6 months, management is conservative (lacrimal sac massage). Subsequently, probing is the first-line invasive treatment [8,10]. The initial probing success rate is 87.2% in children aged 2 weeks to 41 months, but decreases thereafter [11]. If probing is ineffective, silicone tube intubation (STI) is possible; the success rate is 62–100% [12,13,14].

Dacryoendoscopy enables direct CNDO visualization, thus revealing the various pathologies [8]. Dacryoendoscopy allows analysis of membranous obstructions of the Hasner valve, which are the most common cause of NDO; the obstructions have various morphologies. Obstructions at the distal end of the Hasner valve are divided into simple and complicated types. The simple type is a thin blockage that is readily penetrated; the complicated type involves stenosis or fibrosis [14]. However, Nishi et al. reported that only 15.4% of lower nasolacrimal fibroses were revealed by dacryoendoscopy in CNDO children who failed probing [13]. Lacrimal duct mucosal injuries were identified in 69.2% of such children; dacryoendoscopy-assisted incision of the Hasner valve membrane under nasal endoscopic guidance was associated with a high success rate [15]. Although conventional STI (without dacryoendoscopy) also has a high success rate, the various complications include the creation of false passages, punctum erosion, and the formation of pyogenic granulomas [6,13]. Thus, dacryoendoscopy is being increasingly used for lacrimal drainage system (LDS) examination and treatment [13]. Previously, we reported a higher success rate of dacryoendoscopy-assisted STI compared to cases without dacryoendoscopy in patients with primary acquired nasolacrimal duct obstruction [16]. However, few studies have explored whether the use of dacryoendoscopy during STI for CNDO children enhances treatment success. Also, no single-center, single-surgeon study has determined whether dacryoendoscopy improves STI outcomes, which we investigate herein. Therefore, we aimed to identify the best surgical technique. Additionally, we analyzed the internal LDS morphology and that of the Hasner valve at the distal end of the lacrimal duct.

## 2. Materials and Methods

The study and the data collection protocol were approved by the Institutional Review Board of CHA Bundang Medical Center, Seongnam, Republic of Korea (approval no. 2023-09-035, approval date 31 October 2023). The study adhered to the tenets of the Declaration of Helsinki. Informed consent was obtained from all parents or guardians prior to patient enrolment.

### 2.1. Subjects

We retrospectively reviewed the medical records of CNDO patients who underwent STI at the CHA Bundang Medical Center from January 2013 to January 2022. We included children aged ≥ 12 months, or who had failed probing at least once when aged ≤ 12 months. We explained the stepwise treatment for CNDO to the parents of all subjects, and STI was performed for those who consented to surgical treatment. Before the surgery, we explained the purpose, procedure, and potential complications to the parents of all subjects, and informed consent was obtained. All subjects were divided into dacryoendoscopy and non-dacryoendoscopy groups. The conventional group (STI without dacryoendoscopy) included 52 eyes of 41 patients aged 9–117 months treated from January 2013 to July 2016. Among the patients aged < 12 months, one aged 9 months had previously undergone (unsuccessful) probing 3 times, and 38 eyes belonged to subjects aged < 36 months. The dacryoendoscopy-assisted STI group comprised 33 eyes of 28 patients aged 11–123 months treated from August 2016 to January 2022. Two of these patients were <12 months of age (both aged 11 months), and they had both previously undergone unsuccessful probing twice; twenty-three eyes belonged to subjects < 36 months of age. We diagnosed clinical CNDO when epiphora or “sticky eye” was apparent. We excluded children with congenital anomalies such as Down syndrome, as well as those who did not undergo preoperative dacryoscintigraphy (DSG) and those who were lost to follow-up. At the first follow-up visit 1 week post surgery, signs and symptoms were assessed. Surgical success was defined as either the complete resolution of previous signs and symptoms or a successful irrigation test result; follow-up continued for 3 months after extubation. Worsening of signs and symptoms following successful intubation was considered surgical failure.

### 2.2. Transcanalicular Dacryoplasty

From August 2016, we utilized a dacryoendoscope (FT-203F; Fibertech Co., Tokyo, Japan), which possessed the following specifications: probe outer diameter of 0.6 mm, field of view of 70°, 6000 image elements, and an observation depth ranging from 1 to 10 mm. A sheath (Angio catheter, Daewon, Seoul, Republic of Korea) was used to cover and dilate the lumen, with the possibility of movement within the LDS. The nasal endoscope (7208CA; Karl Strotz, Tuttlingen, Germany) had a probe outer diameter of 2.7 mm, a view angle direction of 0°, and 410,000 image elements. All cases were conducted under general anesthesia, and all STI was performed using both dacryoendoscopy and nasal endoscopy. After a subconjunctival injection of 2.0 mL lidocaine 2% with a 30-G needle, we dilated the upper and lower lacrimal punctum with a punctal dilator and inserted the dacryoendoscope through them. The dacryoendoscopy was slowly moved toward the canaliculus, and gently forward to the lacrimal sac. For clear viewing of the lumen, saline was injected through the water channel. On reaching the lacrimal sac, we held the dacryoendoscopy upright and pushed it forward to the lacrimal duct while visually guided by the dacryoendoscopy. Using dacryoendoscopy, we examined the inside of the LDS to determine the level and pattern of obstruction. When the dacryoendoscope passed through the LDS, the nasal endoscope provided the morphological findings of the distal end of the inferior meatus, also known as Hasner’s valve. A bicanalicular silicone tube with a 0.6 mm diameter (YWL84; E&I Tech, Kyungki-do, Republic of Korea) was inserted while under visual guidance. The tube was then retrieved, and both ends were locked and stabilized under the inferior turbinate. All surgical treatments, STI, and dacryoendoscopy examinations were performed by a single surgeon (H.L.).

### 2.3. Classification of Dacryoendoscopic Findings

The endoscopic findings of CNDO were analyzed based on two criteria: findings observed at both the end of the lacrimal duct, including Hasner’s valve and the LDS. First, the morphology of Hasner’s valve was classified into two types: simple and complicated (fibrosis, stenosis). The simple type was defined as a thin blockage of the Hasner’s valve membrane that could be easily perforated without resistance. The complicated type included fibrosis, which was characterized by a dense fibrous blockage at the end of Hasner’s valve, and stenosis, which was defined as a structural narrowing that caused resistance around Hasner’s valve and resulted in a diameter of the end that was smaller than that of the dacryoendoscope (0.6 mm). The presence of a membrane at the Hasner’s valve was defined as cases with very thick membranes that were clearly visible on dacryoendoscopy and nasal endoscopy. Second, throughout the LDS, the pattern of obstruction was classified into two subtypes: the secretory type, such as mucus or dacryolith, and the structural type, such as stenosis or membrane.

### 2.4. StatisticalAnalysis

IBM SPSS software (ver. 23.0; IBM Corp., Armonk, NY, USA) was used for all statistical analyses. Parametric and non-parametric variables were compared using the independent *t*-test and the Mann-Whitney U test, respectively. The paired *t*-test was used to compare findings before and after surgery. A *p*-value < 0.05 was considered statistically significant.

## 3. Results

All patients were divided into non-dacryoendoscopy and dacryoendoscopy groups (groups A and B, respectively) during STI. All patients underwent evaluation of tear secretion, usually before the age of 36 months. The clinical characteristics and demographic data of all subjects are listed in Table 1. The overall success rate was 91.8%, but the rate was significantly higher in group B (97.0%) than in group A (88.5%) (*p* = 0.038). The mean age of groups A and B was 32.7 ± 28.9 and 39.8 ± 30.4 months, respectively (*p* = 0.064). The mean epiphora duration was 25.5 ± 24.0 and 25.7 ± 27.7 months in groups A and B, respectively (*p* = 0.124). Previous lacrimal sac massage was more common in group A than group B (*p* = 0.032); neither the previous probing nor the STI rate (both *p* = 0.124), nor the tube insertion duration (*p* = 0.209), differed between the groups. The follow-up period was 8.1 ± 2.8 months in group A and 8.1 ± 3.0 months in group B (*p* = 0.956).

Dacryoendoscopy revealed all simple Hasner membrane blockages (defined above) (Figure 1a). During nasal endoscopy, the membrane was readily perforated without resistance or bleeding (Figure 1b). In terms of complicated blockages (defined above), a sickle knife was used during endoscopy to remove thick, dense fibrotic membranes (Figure 1c,d). When stenosis was present, the inferior meatus was narrower than the dacryoendoscope (Figure 1e). However, when the scope was moved, the Hasner valve was perforated as easily as the thin membrane of the simple type (Figure 1f). Sometimes, dacryoendoscopy revealed dacryoliths in the lacrimal sac (Figure 1g); these were fragmented under dacryoendoscopic guidance, the fragments were flushed out with saline, and success was confirmed by nasal endoscopy (Figure 1h).

The morphology of the distal end of the lacrimal duct was simple in twenty-four eyes (72.7%) and complicated in seven eyes (21.2%), including four (12.1%) with fibrosis and three (9.1%) with stenosis. Dacryoliths were encountered in two eyes (6.1%) (Figure 2a). Complicated LDS obstructions were found in the canaliculus (43%), lacrimal sac (43%), and lacrimal punctum (14%) (Figure 2b).

The LDS obstructions were classified by the dacryoendoscopic findings. All LDS patterns were associated with either simple or complicated Hasner’s valve obstructions. Sixteen eyes (48.5%) with stenosis exhibited simple obstructions. Among eleven eyes (33.3%) with mucus, eight and three had simple and complicated Hasner’s valve features, respectively. Four eyes (12.1%) with membranes and two (6.1%) with dacryoliths exhibited complicated Hasner’s valve features (Figure 3a). The LDS obstructive levels were as follows: canaliculus, 42.4%; sac, 39.4%; and nasolacrimal duct, 21.2%. The LDS obstructions were structural (n = 18) or secretory (n = 15) (Figure 3b).

LDS dacryoendoscopy revealed Hasner valve membranes in seven eyes (21.2%). All subjects were younger than 36 months and exhibited LDS structural obstructions. Twenty-six eyes (78.8%) without Hasner valve membranes exhibited both secretory and structural LDS obstructions regardless of age (Table 2). The correlation between the presence of a Hasner valve membrane and the LDS obstruction pattern on dacryoendoscopy was significant (*p* = 0.049). During the postoperative follow-up period, there were no complications related to STI, such as nasal hematoma, punctum erosion, and granuloma formation.

## 4. Discussion

We compared CNDO patients who underwent STI without and with dacryoendoscopy. The success rate was significantly higher in the latter group, at 97.0%, even though the rate of prior lacrimal sac massage was higher. The previous probing rate tended to be higher in the group that underwent dacryoendoscopy. Probing can be associated with false passage formation, LDS injury, and bleeding [17]. Thus, dacryoendoscopy would be expected to be more difficult in such patients. However, the better results of our dacryoendoscopy group can be explained by the real-time views of the pathologies; it was possible to visualize the entire LDS, including Hasner valve membranes. We also used nasal endoscopy to directly visualize the inferior nasal meatus; this assisted the localization of anatomical defects. This approach reduces the risks associated with blind intubation, including hemorrhage, nasal mucosal trauma, and iatrogenic false passage creation at the inferior opening of the nasolacrimal duct [18]. However, there was one failure in the dacryoendoscopic group. The symptoms of a 4-year-old boy improved only after STI, but epiphora recurred after extubation. The irrigation test revealed a good flow, indicating a satisfactory functional condition.

Our patients ranged in age from 11 to 123 months (average age = 45.0 ± 34.2 months) and were thus slightly older than the patients in other studies. Gupta et al. performed dacryoendoscopy on patients aged 9–36 months; the success rate was 100% [13]. Heichel et al. performed dacryoendoscopy on patients aged 1–12 months; the success rate was 94.4% [19]. Matsumura et al. reported an overall success rate of 100% in children aged 1–5 years [14]. As in previous studies, our success rate with patients aged < 36 months was 100%. Araz et al. [20] reported a significant difference in the minimum transverse diameter (in the sagittal plane) of the bony nasolacrimal canal duct between children < 2 and >3 years of age. By 36 months of age, the lacrimal system is fully developed, and LDS pathologies become more complex, which is associated with more secretions and mucus. Also, nasal inflammation caused by allergic rhinitis may affect the disease course, and tearing may be attributable to blepharitis, an epiblepharon, keratitis, or foreign bodies [3].

Notably, our dacryoendoscopic CNDO findings were diverse, particularly in terms of the morphology of the distal end of the Hasner valve membrane, as previously reported [21]. Adult NDO levels and patterns vary to a greater extent than those of children because the causes of NDO are more diverse in adults. For example, the lacrimal duct is the most common site of adult obstruction, but this is not the case in children who have CNDO. However, even in children, it is essential to evaluate both the LDS and the Hasner valve when exploring NDO via both dacryoendoscopy and nasal endoscopy; the conventional blind intubation technique is no longer considered appropriate.

In terms of the pathological LDS changes, of the twenty eyes in this study exhibiting stenosis and membranes, sixteen were simple and four were complicated. Notably, all patients with stenosis were in the simple group, and all patients with membranes were in the complicated group. Of thirteen eyes with secretory patterns (mucus and stones), eight were simple and five were complicated. Thus, when the LDS is narrow or stenotic, as revealed by dacryoendoscopy, the Hasner valve membrane can easily be perforated provided LDS damage is carefully avoided. However, if the LDS exhibits a secretory pathology, the Hasner valve membrane may be complicated to deal with. It is essential to treat a fibrotic or stenotic membrane at the end of the NLD opening to ensure a favorable outcome.

This study had several limitations. It was a retrospective single-institution study with a small sample size, and further studies are needed. Also, all of the children were Korean; generalization of our findings to other populations should be carried out with caution. Finally, it was difficult to compare the CNDO types between the two groups because data on the Hasner valve and LDS obstructions in the conventional group were lacking. Future research will be needed to elucidate the pathological mechanisms causing changes within the LDS in CNDO.

In conclusion, STI combined with transcanalicular dacryoplasty was more successful than the conventional technique (without dacryoendoscopy). The CNDO etiopathologies identified by dacryoendoscopy were diverse, not only at the Hasner valve membrane but also throughout the LDS. Dacryoendoscopy-based direct visualization and recanalization effectively improve the STI outcomes of CNDO patients, particularly those younger than 36 months.

## Figures and Tables

**Figure 1 jcm-12-07370-f001:**
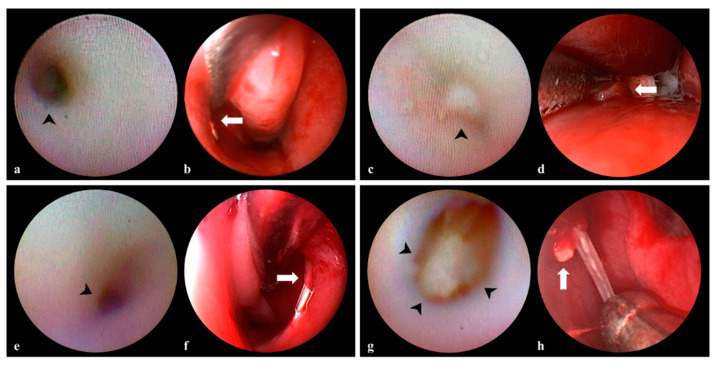
Classification of dacryoendoscopic findings (**a**,**c**,**e**,**g**) with nasoendoscopic findings (**b**,**d**,**f**,**h**) at the distal end of the nasolacrimal duct in patients with congenital nasolacrimal duct obstruction. Simple type presented, the normal lacrimal duct (arrowhead) in the right eye (**a**) and easily perforated at Hasner’s valve (arrow) (**b**). Complicated type demonstrated dense fibrosis in the left lacrimal duct (arrowhead) (**c**) and thick fibrous membrane was removed using a sickle knife (arrow) (**d**). Stenosis type was noted before the left inferior meatus (arrowhead) (**e**) and thin membrane was readily perforated at Hasner’s valve (arrow) (**f**). Dacryolith was noted at the right lacrimal sac (arrowhead) (**g**) and fragments of dacryolith were removed (arrow) (**h**).

**Figure 2 jcm-12-07370-f002:**
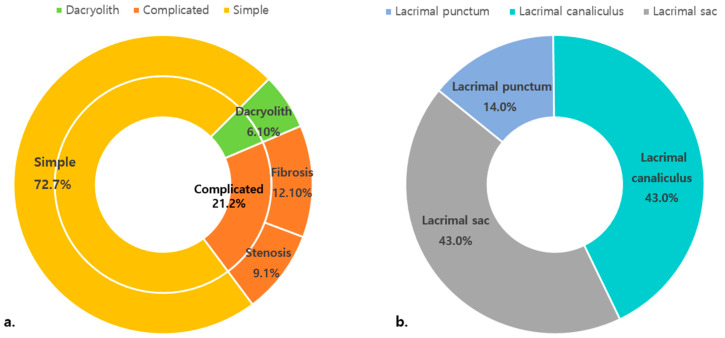
Distribution of the patients with congenital nasolacrimal duct obstruction. (**a**) Type of obstruction at the end of nasolacrimal duct by dacryoendoscopy (n = 33). (**b**) Level of obstruction in the lacrimal drainage system in the complicated type (n = 7).

**Figure 3 jcm-12-07370-f003:**
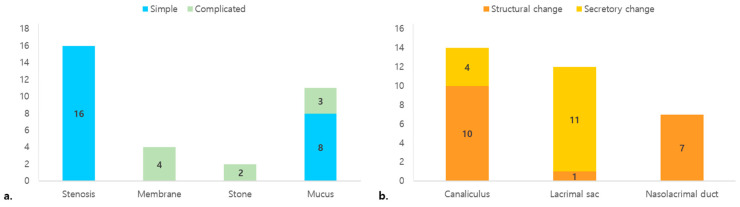
Correlation of dacryoendoscopic findings of the lacrimal drainage system and the type of congenital nasolacrimal duct obstruction. (**a**) Pattern of obstruction in the lacrimal drainage system according to the type of obstruction at the distal end of the nasolacrimal duct. (**b**) Level of obstruction in the lacrimal drainage system according to the pattern of obstruction.

**Table 1 jcm-12-07370-t001:** Clinical characteristics and demography in the patients according to the use of dacryoendoscopy.

	Group A (without Dacryoendoscopy)			Group B (with Dacryoendoscopy)			*p*
	≤36 Months	>36 Months	Total	≤36 Months	>36 Months	Total	
Age (month)	21.4 ± 7.3	63.5 ± 41.5	32.7 ± 28.9	22.6 ± 8.5	79.4 ± 24.9	39.8 ± 30.4	0.064
Patient-level characteristics	N = 30	N = 11	N = 41	N = 19	N = 9	N = 28	
Sex (male:female)	18:12	2:9	20:21	10:9	6:3	16:12	0.349
Epiphora duration (months)	18.1 ± 8.5	45.6 ± 38.1	25.5 ± 24.0	21.5 ± 9.0	47.1 ± 38.3	25.7 ± 27.7	0.124
Eye-level characteristics	N = 38	N = 14	N = 52	N = 23	N = 10	N = 33	
Laterality							
Unilateral (%)	24 (63.2)	8 (57.1)	32 (61.5)	17 (73.9)	8 (80.0)	25 (75.8)	0.174
Bilateral (%)	14 (36.8)	6 (42.9)	20 (38.5)	6 (26.1)	2 (20.0)	8 (24.2)	
Previous massage							
Yes (%)	5 (13.2)	0 (0.0)	5 (9.6)	6 (26.1)	3 (30.0)	9 (27.3)	0.032 *
No (%)	33 (86.8)	14 (100.0)	47 (90.4)	17 (73.9)	7 (70.0)	24 (72.7)	
Previous probing or STI							
Yes (%)	1 (2.6)	5 (35.7)	6 (11.5)	5 (21.7)	3 (30.0)	8 (24.2)	0.124
No (%)	37 (97.4)	9 (64.3)	46 (88.5)	18 (78.3)	7 (70.0)	25 (75.8)	
Duration of tube insertion (months)	5.3 ± 0.9	5.1 ± 1.9	5.2 ± 1.7	4.7 ± 3.4	4.2 ± 1.2	4.5 ± 2.8	0.209
Follow-up period (months)	8.1 ± 2.8	8.0 ± 2.9	8.1 ± 2.8	7.5 ± 2.9	8.1 ± 2.7	7.7 ± 2.6	0.426
Success rate (%)	32 (84.2)	14 (100.0)	46 (88.5)	23 (100.0)	9 (90.0)	32 (97.0)	0.038 *

STI: silicone tube intubation, *: *p* < 0.05.

**Table 2 jcm-12-07370-t002:** Dacryoendoscopic findings in the lacrimal drainage system associated with the membrane at the Hasner’s valve.

	Present (n = 7)				Absent (n = 26)				Total (n = 33)	*p*
	≤36 Months	>36 Months	Total	*p*	≤36 Months	>36 Months	Total	*p*		
Secretory change	0 (0.0)	0 (0.0)	0 (0.0)	NC	8 (50.0)	5 (50.0)	13 (50.0)	1.000	13 (37.1)	
Structural change	7 (100.0)	0 (0.0)	7 (100.0)		8 (50.0)	5 (50.0)	13 (50.0)		20 (62.9)	
Total	7 (100.0)	0 (0.0)	7 (100.0)		16 (61.5)	10 (38.5)	26 (100.0)		33 (100.0)	0.049 *

## Data Availability

The datasets analyzed during the current study are available from the corresponding author on reasonable requests.

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
