# Peer review of "Comparison of Outcomes of Silicone Tube Intubation with or without Dacryoendoscopy for the Treatment of Congenital Nasolacrimal Duct Obstruction"

_jcm, 2023, doi:10.3390/jcm12237370_

Round 1
Reviewer 1 Report (Previous Reviewer 1)
Comments and Suggestions for Authors
I would like to thank the authors for addressing all of my comments and especially for sending the manuscript to professional English editing service. The manuscript is improved significantly and reads very well now. I also read reviewer #2’s comments, and it appears that the authors did their best to incorporate reviewer #2’s suggestions.
Author Response
Please see the attachment.

Reviewer 2 Report (Previous Reviewer 2)
Comments and Suggestions for Authors
Thank you for the corrections on the manuscript of this relevant study. One issue which has been highlighted and should be emphasised again, is the fact that the authors document/advocate in L28 that the first-line non-surgical treatment consists of lacrimal massage. Table 1 (fifth line), however, reveals that most patients did not receive the recommended first-line treatment. This discrepancy requires to be addressed despite the retrospective data used. It could be claimed that the authors did not follow treatment guidelines in their surgical practice (despite self-declaration in the introduction). While unlikely, in case of a litigation issue this could be exploited (wrong or overtreatment). Thus, I would recommend to the author to state that no over- or false treatment and/ or complications have occurred.
Comments on the Quality of English LanguageWell-written article.
Author Response
Please see the attachment.

This manuscript is a resubmission of an earlier submission. The following is a list of the peer review reports and author responses from that submission.
Round 1
Reviewer 1 Report
Comments and Suggestions for Authors
Overview of the paper:
The paper focuses on the study of congenital nasolacrimal duct obstruction (CNDO) treatment, comparing the efficacy of two methods. The research seems to lean heavily on the results obtained, especially concerning the dacryoendoscopic method.
Strengths:
1. Comprehensiveness: The paper offers a thorough investigation into the topic and provides valuable data, especially on the potential advantages of dacryoendoscopy in CNDO treatment.
2. Relevance: The subject is pertinent to the field of ophthalmology, with the potential to influence clinical practices concerning CNDO treatment.
3. Detail: The Materials and Methods section is comprehensive, outlining the study's setting, subjects, procedure, and statistical analysis.
Major issues:
Abstract:
· Objectives missing: The objective of the study isn't clearly stated. It's assumed from the content that the aim is to compare success rates between two methods, but this should be explicitly stated.
Minor issues:
Abstract:
· Typo: The last sentence has a typo. "Varing" should be "varying".
Introduction:
· Redundancy: The term "lacrimal drainage system (LDS)" is introduced with its abbreviation twice. Please remove the second instance of the term and keep the abbreviation only.
Results:
· Figures 2 and 3 are well created with beautiful colors chosen. However, the figures are blurry. Kindly use a higher definition copy for better legibility.
· There are several grammatical errors and awkward phrasings that need correction. For example:
o “Each group was analyzed based on the age of 36 months by which tear secretion function would be completed” to “Each group was analyzed based on tear secretion function, which is typically completed by the age of 36 months”
o "Previous lacrimal sac massage history was higher in the Group A than Group B (p=0.032)”. Remove “the” before Group A
o and There was no difference...". The word "There" should not be capitalized.
o "The others of 26 eyes (78.8%) without the Hasner’s valve membrane demonstrated...". The phrase "The others of 26 eyes" is awkward and could be rephrased for clarity.
· Statements like "which was smaller than others (Fig. 2b)" could benefit from more specificity. What does "others" refer to in this context? Please clarify.
· The paper uses terms like "simple type," "complicated type," and various subtypes without clear prior definition or context. This can be confusing for readers unfamiliar with these classifications. A sentence or two to define what simple CNDO vs. complicated (senotic and fibrotic) would solve this issue.
Discussion:
· Grammatical errors:
o “which appears to statistically significantly higher than conventional STI,” lacks a verb. Must add “be” before “statistically.”
o “there were 1 cases of failure”ïƒ there was 1 case of failure.
Comments on the Quality of English Language
I highly advise professional editing of the paper. There are several instances in the results and discussion that could benefit from English editing due to the presence of several grammatical errors and awkward phrasing.
Reviewer 2 Report
Comments and Suggestions for Authors
The study “Comparison of Outcomes of Silicone Tube Intubation with or without Dacryoendoscopy for the Treatment of Congenital Nasolacrimal Duct Obstruction” assesses patients undergoing silicone tube intubation (STI) for congenital nasolacrimal duct obstruction (CNDO) with and without dacryoendoscopy on 85 eyes. The article includes an abstract, 3 keywords, the sections “Introduction”, “Material and Methods”, “Results” and “conclusion”, 2 tables, 3 figures, the formal statements (missing informed consent statement) and 20 references.
The study has to be reconsidered for the following reasons.
The recommended treatment for CNDO remains a controversy which becomes clear when evaluating references [4-8] which come to different conclusions for an ideal treatment. In addition, it is not always clear if epidemiological reflections are evaluated in combination with clinical reasoning which is necessary to produce a conclusive and valid treatment guideline.
Especially, it remains unclear what the spontaneous resolution rate is (over long term) and under what conditions, less invasive procedures (massage, probing) should be performed. It would be ideal to have a natural history study over a longer period of time or alternatively a cohort study with longer observation time. This would allow to further characterise the course of the condition and potentially allow to define subgroups for stratification of treatment strategies. The present study compares two different surgical approaches and thus suggests a surgical approach as a treatment. While a procedure with dacryoendoscopy may be more effective according to the analysis of the authors, conservative strategies should not be neglected. The present study reveals that, according to Table 1, many patients did not receive a massage or a less invasive treatment, which could be criticised. In addition, complications are not reported in the present study, which is an essential component of a clinical study evaluating a surgical procedure.
If the authors aim to advocate for a surgical treatment strategy with the purpose of changing the treatment guidelines, a more thorough analysis should be performed (see above). An argument for a possible change of guidelines could be the increased risk/prevalence of anisometropic amblyopia, although the rates are still relatively low. If the purpose of the study is to report data from a retrospective review of cases operated with and without dacryoendoscopy, it would be recommended to present the results without entering a discussion around controversial treatment approaches and clearly state that the article aims to identify the best surgical technique.
Due to the fact that litigation issues could arise from the article, I would advise to revise and refocus the study.